# The Epidemiology of *Entamoeba histolytica* Infection and Its Associated Risk Factors among Domestic and Imported Patients in Taiwan during the 2011–2020 Period

**DOI:** 10.3390/medicina58060820

**Published:** 2022-06-17

**Authors:** Fu-Huang Lin, Bao-Chung Chen, Yu-Ching Chou, Wu-Chien Chien, Chi-Hsiang Chung, Chi-Jeng Hsieh, Chia-Peng Yu

**Affiliations:** 1School of Public Health, National Defense Medical Center, Taipei City 11490, Taiwan; noldling@ms10.hinet.net (F.-H.L.); trishow@mail.ndmctsgh.edu.tw (Y.-C.C.); chienwu@mail.ndmctsgh.edu.tw (W.-C.C.); g694810042@gmail.com (C.-H.C.); 2Division of Gastroenterology, Department of Internal Medicine, Tri-Service General Hospital, National Defense Medical Center, Taipei City 11490, Taiwan; staineely@yahoo.com.tw; 3Department of Medical Research, Tri-Service General Hospital, National Defense Medical Center, Taipei City 11490, Taiwan; 4Graduate Institute of Life Sciences, National Defense Medical Center, Taipei City 11490, Taiwan; 5Taiwanese Injury Prevention and Safety Promotion Association, Taipei City 11490, Taiwan; 6Department of Health Care Administration, Asia Eastern University of Science and Technology, New Taipei City 22061, Taiwan; fl004@mail.aeust.edu.tw

**Keywords:** diarrhea, epidemiology, amebiasis, *Entamoeba histolytica*, retrospective study

## Abstract

*Background and Objectives*: Amebiasis remains an important public health problem worldwide, and immigration and increased international travel have affected incident disease cases. This study assesses the prevalence of *Entamoeba histolytica* in Taiwan between 2011 and 2020 by analyzing data from surveillance programs conducted by the Centers for Disease Control of Taiwan (TCDC) on laboratory-confirmed cases. *Materials and Methods*: The *E**. histolytica* infection-related data reported to the National Infectious Diseases Statistics System at the TCDC from 1 January 2011 to 31 December 2020 were collected, including age, gender, place of residence, and the geographic season of exposure for each case. *Results*: In total, 3066 cases with *E. histolytica* infections were included in our analysis. Among them, 1735 (57%) cases were imported, and 1331 (43%) were locally acquired. The average annual incidence rate of *E. histolytica* infections in Taiwan between 2011 and 2020 was 10.6 and 16.1 per 1,000,000 patients. There were statistical differences in gender, age group, and place of residence (*p* < 0.001) by the source distribution of cases. Also, these differences were found every year (*p* < 0.05). There were statistical differences in gender and age group (*p* < 0.001) by place of residence (*p* < 0.001). The only difference between the distribution of cases and age group was in gender (*p* < 0.001). Eight patients with amebiasis died, and the fatality rate was 0.3% (8/3066), of whom 75% (6/8) were male, and 75% (6/8) were over 45 years old. This study demonstrates that multiple linear regression analysis shows positive associations between NO_2_ concentration and amebiasis cases (B value = 2.569, *p* = 0.019), O_3_ concentration and amebiasis cases (B value = 0.294, *p* = 0.008), and temperature and amebiasis cases (B value = 1.096, *p* = 0.046). *Conclusions:* This study is the first report of confirmed *E. histolytica* cases from TCDC surveillance data between 2011 and 2020. This study showed the importance of long periods, air pollutants, and geographically comprehensive analysis for estimating the effect of amebiasis transmission in Taiwan’s populations.

## 1. Introduction

Amebiasis is an infectious disease caused by *Entamoeba histolytica* [1]. Each year, amebiasis affects an estimated 50 million people worldwide and causes 100,000 deaths [2]. *E. histolytica* can cause invasive intestinal and extraintestinal diseases [3]. It is mainly parasitic in the intestinal tract, and most infected people have no apparent symptoms. This protozoan may invade the intestinal wall tissue of the host, causing intestinal symptoms, abdominal discomfort, intermittent dysentery or constipation, accompanied by fever, shivering, bloody stool, or mucousy soft stool. Symptoms last for approximately one to three weeks. However, extraintestinal infections, such as liver abscesses, lung abscesses, or brain abscesses, may develop in a few people, leading to severe and potentially fatal risks [4,5].

*E. histolytica* is associated with contaminated food or sewage, and it is transmitted by the fecal–oral route [6]. The growth of *E. histolytica* can be divided into two stages. The cyst of *E. histolytica* can survive outside the host in a low temperature and harsh environment for several months. The infection can occur when the person puts things into their mouth that have touched the feces of the infected person with *E. histolytica*. After the cyst are swallowed, they become active in the intestinal tract and reproduce to produce the next generation. Amebiasis has a worldwide distribution, but the infection is more common in tropical and subtropical developing countries. Infections are prone to occur in orphanages, correctional institutions, shelters, and prisons, with people living in groups. Investigations in Taiwan have found that the infection is relatively common in correctional institutions and mentally ill shelters, and it is closely related to living habits, sanitary conditions, environmental pollution, and group living conditions. In addition, high risk groups are men who have sex with men, individuals from epidemic areas, foreign workers and immigrants from epidemic areas, and inmate populations [7,8]. Amebiasis is a zoonotic infectious disease. *E. histolytica* is a common human pathogen, and it also occurs in non-human primates. Other animals, such as dogs, cats, pigs, cows, and mice are less infected [9].

Taiwan is a developed, island-type country with a moist subtropical climate, except for the tropical southern part. The monthly average of temperature is 16 °C to 29 °C, and the relative humidity is 75% to 90%. The climatic conditions in Taiwan are suitable for the survival and the spread of amebiasis, resulting in approximately 100 local cases in Taiwan every year [10], a decrease in the quality of life of patients, threats to life and health, and the consumption of medical resources. Nevertheless, few studies reported the epidemiological characteristics of amebiasis-related infection in Taiwan. This study assesses the occurrence of amebiasis infections and their effect on environmental factors, and it examines regional and demographic associations.

## 2. Materials and Methods

### 2.1. Ethical Policy

The data came from the Taiwan Centers for Disease and Control (TCDC), and it is public health surveillance rather than research that involved human subjects. Thus, institutional review board approval and informed consent were not required for these deidentified secondary data analyses [11,12,13]. 

### 2.2. Data Source

The geographical coordinates of Taiwan are 23°4 North and 121°0 East, and it has a population of approximately 23 million on a land area of 36,188 km^2^ for a population density of 627/km^2^. The National Infectious Diseases Statistics System (NIDSS) has reported amebiasis cases to the Centers for Disease Control of Taiwan (TCDC) from 1990. As amebiasis infection is a reportable disease in Taiwan, physicians must report all cases by entering the data into local databases and electronically forwarding the data to the TCDC within 24 h of case confirmation, using TCDC-developed software [14]. Over 84% of physicians reported the notifiable disease to the TCDC after the diagnosis [15]. After receiving the report, a TCDC-assigned epidemiological team (parasitologist, field epidemiologist, and public health nurse) follows the patient, verifies the diagnosis, and collects patient information. Investigations included face-to-face interviews, telephone calls and correspondence with healthcare providers, and interviews with *E. histolytica*-infected patients.

The *E. histolytica* infection-related data reported to the NIDSS at the TCDC from 1 January 2011 to 31 December 2020 were collected, including age, gender, place of residence, and geographic season of exposure for each case [10].

### 2.3. Case Definition

This study refers to the TCDC guidelines for patients with amebiasis infection [16], and it defines the clinical, laboratory, and epidemiological conditions of the case. The clinical conditions are mild, chronic-to-severe diarrhea, mucus and blood filaments in feces, acute and severe diarrhea, intermittent diarrhea, abdominal pain, fever, nausea, and vomiting. However, most *E. histolytica* carriers have no obvious clinical symptoms. The test must meet one of the following conditions: (1) Clinical specimens (feces, tissue or abscess extracts, and other types) are confirmed to be *E. histolytica* through the molecular biology nucleic acid test of the TCDC laboratory. (2) Intestinal and extraintestinal tissue slices or ulcers were cured, and active *E. histolytica* was found. (3) The case has clinical symptoms (fever or right upper quadrant pain). It is diagnosed as a liver abscess by ultrasound or computed tomography, and the serological test is positive for the antibody of *E. histolytica*. Epidemiological requirements must meet one of the following conditions: (1) Had close contact with a confirmed case. (2) Consumed food and drinking water that has been directly or indirectly contaminated by feces of patients or carriers. Furthermore, the “definition of a notified case” refers to one of the following conditions: (1) It meets the clinical conditions. (2) It meets the inspection conditions after testing by a medical laboratory. The “definition of a confirmed case” refers to one of the following conditions: (1) The first item of the test conditions is met. (2) It meets the clinical conditions (fever or right upper quadrant pain) and the second or third test conditions.

### 2.4. Laboratory Examination 

The stool and the blood samples were collected from patients suspected of having *E. histolytica* infection [17]. For the Entamoeba spp, the microscopic examination was screen in various hospital laboratories. To confirm the diagnosis and to identify the amebic species, positive specimens were sent to the TCDC laboratory [18]. The antibody detection methods were tested with IHA (Dade Behring Diagnostics, Marburg, Germany) and TechLab (Blacksburg, VA, USA) [19,20]. DNA-based diagnostic tests were based on the amplification of the small-subunit rRNA gene of *E. histolytica*, *E. dispar* [21,22]. 

### 2.5. Surveillance of Environmental Factors

This study analyzes the monthly data of air pollutants provided by the air quality monitoring network of the Environmental Protection Agency from 2011 to 2020 [23], including total suspended particulates (TSP), particulate matter 2.5 (PM2.5), nitrogen dioxide (NO_2_), sulfur dioxide, carbon monoxide (CO), and ozone (O_3_). The monthly data of weather factors (temperature, rainfall, relative humidity, atmospheric pressure, rainfall days, and sunshine hours) provided by the Meteorological Bureau of the Ministry of Communications from 2011 to 2020 were analyzed [24]. Statistical analysis and correlation tests were used to understand the temporal and the spatial variation trend of air pollutants and meteorological factors and their correlation with amebiasis case numbers.

### 2.6. Statistical Analysis

We identified amebiasis cases from 2011 to 2020 from the database, and we examined their epidemiological differences and trends. The continuous variables were described as means and standard deviations. Categorical data was analyzed by the chi-square test. This study also computed the odds ratio by logistic regression and the 95% confidence interval in the parameter estimation. All analyses were performed using the Statistical Package for the Social Sciences software software (IBM SPSS Statistics 21; Asia Analytics Taiwan Ltd., Taipei, Taiwan). Statistical significance was defined as a *p*-value of <0.05.

## 3. Results

### 3.1. Surveillance

Figure 1 presents the amebiasis importation rate and the number of domestic and imported cases in Taiwan by year from 1 January 2011 through 31 December 2020. The annual case numbers ranged from 255 to 379. The importation per 1,000,000 inbound travelers was 22.2 in 2011 and 92.9 in 2020. In addition, during the study period, the annual incidence of amebiasis, male and female cases, cases in each age group, cases in different seasons, and cases in different residences are shown in Figure 2.

### 3.2. Outcome of Epidemiological Features

Confirmed *E. histolytica* infection cases were 3066 for which data related to the risk of infection (sex, age group, and area of residence) with their statistical significances were obtained (Table 1). A comparison of the risk factors and cases each year revealed the following findings: (1) No significant differences for the season were observed; (2) For cases of style, sex, age group, and residency, all epidemiological features were statistically significant (*p* < 0.05) (Table 2). A comparison of the risk factors and cases of the season revealed the following findings: No significant differences in sex, age group, and residency were observed (Table 3). A comparison of the risk factors and the cases of residency revealed the following findings: For sex and age group, all epidemiological features were statistically significant (*p* < 0.05) (Table 4). A comparison of the risk factors and cases of sex revealed the following findings: No significant differences for age group were observed (Table 5).

A total of 1735 confirmed cases of *E. histolytica* were imported from abroad during 2011–2020. The purpose of entering Taiwan was mainly for tourism and business (including foreign workers). The country with the most cases of infection was Indonesia, with 1385 cases (Table 6) of which the number of cases in 2014 was the highest in the past seven years (197 cases). The annual import rate is shown in Figure 1, and the annual distribution of imported cases is shown in Figure 2.

Travel destinations of 1,695 imported cases of *E. histolytica* infection in Taiwan (only countries with at least 10 cases of infection with *E. histolytica* were listed). A total of 1385 (79.8%) had been to Indonesia, 183 (10.5%) to the Philippines, 51 (2.9%) to Vietnam, and 25 (1.4%) to Thailand. The relative risk (RR) of Amebiasis for travelers from Taiwan to Indonesia was 544.0, to the Philippines it was 60.7, to India it was 25.8, to Vietnam it was 16.0, and to Thailand it was 9.3 when compared with traveling to China (Table 7).

During the 10-year investigation period, there were eight death cases due to amebiasis: six males (75%) and two females (25%); two individuals were 20–44 years old, three individuals were 45–64 years old, and three individuals ≥65 years old (Table 8).

### 3.3. Outcome of Environmental Features

Air pollution factors were associated with amebiasis cases by multiple linear regression analysis. The value was R^2^ = 0.145, F =2.519 (*p* = 0.027, df = 6, 89). As shown in Table 9: The B value of the non-standardization coefficient is 2.569, standard error is 1.074, and the *p* value is 0.019 for NO_2_ pollutants. The B value of the non-standardization coefficient is 0.294, standard error is 0.109, and the *p* value is 0.008 for O_3_ pollutants. Furthermore, climate factors were associated with amebiasis cases by multiple linear regression analysis. The value was R^2^ = 0.073, F = 1.165 (*p* = 0.332, df = 6, 89). As shown in Table 10: The B value of the non-standardization coefficient is 1.096, standard error is 0.542, and *p* value is 0.046 for the temperature factor. 

## 4. Discussion

From 2011 to 2020, there were 3066 confirmed cases of *E. histolytica* infections in Taiwan: 43% were imported and 57% were indigenous, similar to a previous study [17]. Epidemiology uses demographic methods to characterize the distribution of a particular organism and analyzes the data to ascertain the determinants of that particular distribution [25]. Epidemiology also explores disease distribution and disease occurrence determinants in populations in time and in space and the epidemiological characteristics associated with disease transmission, presentation, and outcome. This discipline has always been driven by policy interventions or disease prevention measures [2,26].

The high incidence rate of men in local cases is the opposite of women in foreign countries. It is inferred that the reason may be different national conditions or different personal hygiene habits. Second, most imported cases are 20–29 years old, whereas most local cases are 30–39 years old. Imported cases are concentrated in summer and fall, whereas local cases are concentrated in spring and fall. The similarity between imported and local cases is that both are concentrated in the northern region. Taiwan’s government can use these similarities and differences in epidemiological characteristics as the basis for implementing its epidemic prevention policy plan or strategy. 

Among all epidemiological characteristics, seasonal variation is the least affected factor by the distribution of all confirmed cases. Namely, seasonal variation is insufficient to affect the significant increase or decrease of confirmed cases, indicating that the disease in South Asia, Southeast Asian countries, or even Taiwan is endemic. In other words, there are always traces of *E. histolytica* in Taiwan that threaten public health and increase the clinical medical burden. At this stage, all of these major issues need to be overcome, prevented, and controlled. The Taiwan government health department should actively propose local prevention strategies, oversea border control, and implement surveillance operations to control the epidemic effectively and to reduce the number of cases so as to eliminate the health threat.

Humans swallow *E. histolytica* by ingesting water or food or having unclean hands contaminated with *E. histolytica* viable cysts. Foodborne exposure infection is the most common route of transmission, especially when food handlers do not clean their hands after using the toilet or their feces become agricultural fertilizer. Some clinicians will use irrigation devices to obtain infection during rectal enema in rare cases. In developing countries, such as India and Mexico, large-scale epidemics of waterborne *E. histolytica* infection often occur [26]. In developed countries, *E. histolytica* infection is common among travelers, new immigrants, gay men, and prison inmates [27,28], similar to the findings of this study. The transferred cases of *E. histolytica* from outside the epidemic area showed a high incidence.

According to data from studies in other countries, about half of travelers with enteric *E. histolytica* infection who return to developed countries are from Asia and Africa [29], similar to the results of this study. The increase in international passenger flows has affected the incidence. This study showed the incidence of *E. histolytica* infection in Taiwan increased with the increased incidence of *E. histolytica* among inbound tourists.

This study confirmed the risk of contracting *E. histolytica* during travel to countries where the disease is endemic. Infections with travel-related enteric pathogens occur during travel and after the journey [30]. Some studies have stated that about 8% of international travelers are seriously ill during or after travel and require medical care [31]. Travelers contribute to the global spread of infectious diseases, including novel and emerging pathogens. Therefore, monitoring travel-related morbidity is an integral part of global public health monitoring. It will become increasingly important with the expansion of global international flights and the growth of the travel population [32]. It is necessary to establish a sensitive, rapid, and accurate surveillance system and to implement an amebiasis screening program for inbound travelers (especially immigrants/foreign workers) from epidemic areas to control amebiasis infection in Taiwan. 

According to the newly revised nomenclature, *E. histolytica* is the pathogenic strain, and the non-pathogenic strain is *E. dispar* [33]. The two species can be distinguished by various experimental methods, but they cannot be distinguished morphologically [34,35]. Terms such as molecular epidemiology often appear in the infectious disease literature, including the study of molecular typing of pathogenic strains of infectious factors (i.e., pathogens). DNA technology has made an outstanding contribution to the study of amebiasis infection in humans. Biomarkers provide new opportunities to overcome some limitations of traditional scientific methods. However, these biomarkers should be verified in molecular epidemiological studies on bias and confounders [36]. Genetic diversity and kinship conducted by previous studies have confirmed several *E. dispar* genotypes, which may be related to intestinal or liver tissue damage. They are similar to the symptoms caused by *E. histolytica* [37]. The intestinal symptoms may also be caused by *E. dispar* infection [38]. Furthermore, *E. dispar* infection can often be as a fecal–oral infection indicator. According to previous studies by Taiwanese scholars, the cases cannot rule out the possibility of infection by other pathogens (e.g., *E. dispar*), mainly because *E. histolytica* in Taiwan is a legal infectious disease, and its applied experimental diagnostic methods do not cover all possible intestinal infection categories [17].

In the absence of a policy for the impact of climate change on public health, many countries are developing and implementing response measures for climate change adaptation [38]. Climate disasters in 2020, such as wildfires, hurricanes, and heatwaves, show that climate change has an increasing effect on health, and it will undoubtedly seriously threaten human health. In addition, COVID-19 is a great pandemic. We know that only by preventing a foreseeable public security crisis can we avoid unnecessary illness and death. Studies have shown that global warming may lead to more deaths from heatwaves and the increased incidence and mortality from waterborne diseases, such as foodborne diseases [39]. A previous study indicated that well-known socioeconomic factors contributing to the incidence of *E. histolytica* found that temperature and precipitation were associated with a higher risk of infection [40]. This study also confirmed that when the temperature increased, the number of *E. histolytica* cases also increased, and there was a positive correlation between them. These results are similar to those of other studies. To our knowledge, this is the first study to find that when the concentration of the air pollutants O_3_ and NO_2_ increased, the number of amebiasis cases also increased, and there was a positive correlation. Our study inferred that the possible reasons might be the high concentration of O_3_ in the fall and the high concentration of NO_2_ in the late fall and the early winter, which are prone season(s) for *E. histolytica*. It was estimated that there was a correlation between these air pollutants and the disease. Therefore, this study suggests that Taiwan’s official policy should closely monitor local climate factors and air pollution concentration changes. Once the environmental data changes slightly or fluctuates violently, the media and the public should be informed. People can quickly respond and follow up to reduce environmental pressures and the threat to public health.

## 5. Limitations

This study has two strengths. First, it analyzed the laboratory-confirmed cases of amebiasis and then combined confirmed cases with the number of inbound passengers from epidemic countries and data on overseas cases. These data enabled us to analyze the risk of tourists to areas where amebiasis is endemic. This is the first study based on the database compiled by the infrastructure of the Taiwan Tourism Bureau. This study determined the most representative countries of origin and the number of inbound passengers and tried to perform a limited analysis and comparison. This study has two methodological limitations. First, both local and overseas cases are based on surveillance data. Therefore, reporting bias may affect the number of cases. This bias can occur anywhere in the reporting chain—from the patient seeking medical care to the medical care registry recording and confirming cases. Some studies have stated that many notifiable infectious diseases may be underestimated [41]. Some studies indicated that at least 10% of asymptomatic patients infected with pathogenic strains developed amebic colitis [2]. Asymptomatic ameba infection cases may have been in Taiwan during the study period, but they were not collected in the analysis. Our study may underestimate the cases. Second, this study cannot test the impact of individual susceptibility. Therefore, investigating the specific role of other risk factors in the transmission of *E. histolytica* infection is very important.

## 6. Conclusions

In conclusion, there is a high risk of contracting *E. histolytica* for travelers to South and Southeast Asia. It is necessary to establish a sensitive, rapid, and accurate epidemic monitoring system and to implement a screening plan for *E. histolytica* among inbound passengers/foreign workers from epidemic countries to effectively control *E. histolytica* infection in Taiwan. In addition, strategies to improve the environment and to maintain the overall ecology should be proposed for the active prevention and control of *E. histolytica* and to monitor and treat air pollutant concentration, temperature, rainfall, and other climate change factors in Taiwan.

## Figures and Tables

**Figure 1 medicina-58-00820-f001:**
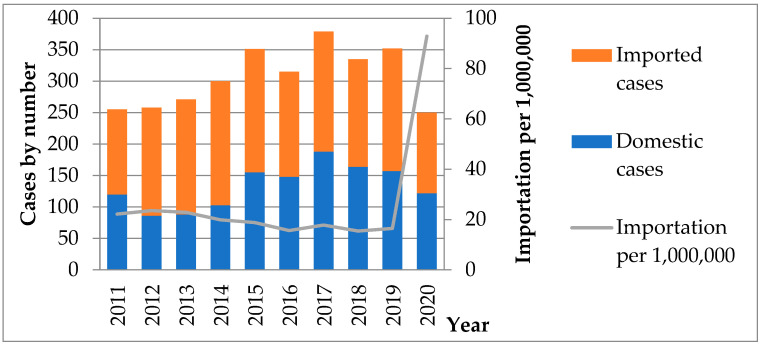
Amebiasis importation rate and the number of domestic and imported cases in Taiwan by year.

**Figure 2 medicina-58-00820-f002:**
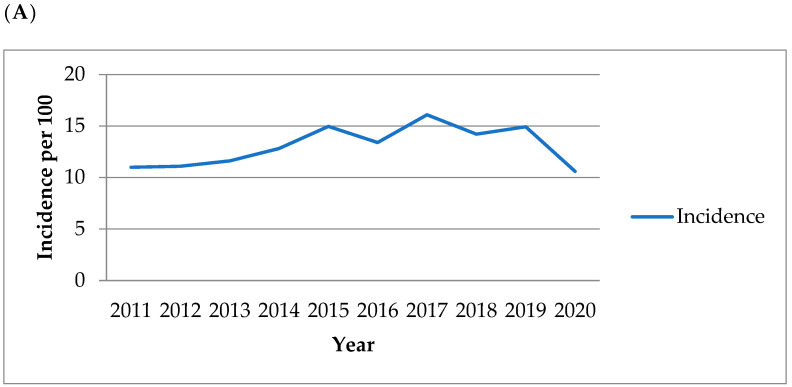
Incidence of confirmed amebiasis among patients in Taiwan according to (**A**) population, (**B**) sex, (**C**) age group, (**D**) season, and (**E**) residency by year from 2011 to 2020.

**Table 1 medicina-58-00820-t001:** The characteristics of domestic and imported cases of amebiasis in Taiwan from 2011 and 2020.

Variables	All Cases		Domestic Cases		Imported Cases		*p*
	N = 3066, %		N = 1331, %		N = 1735, %		
Sex							
Male	1707	55.7	1085	81.5	622	35.9	<0.001
Female	1359	44.3	246	18.5	1113	64.1	
Age							
<20	26	0.8	10	0.8	16	0.9	<0.001
20–29	1075	35.1	283	21.3	792	45.6	
30–39	1185	38.6	450	33.8	735	42.4	
40–49	408	13.3	252	18.9	156	9.0	
50–59	186	6.1	164	12.3	22	1.3	
≥60	186	6.1	172	12.9	14	0.8	
Season							
Spring	773	25.2	346	26.0	427	24.6	0.251
Summer	785	25.6	321	24.1	464	26.7	
Fall	808	26.4	345	25.9	463	26.7	
Winter	700	22.8	319	24.0	381	22.0	
Residency							
Taipei area	1055	34.4	516	38.8	539	31.1	<0.001
Northern	410	13.4	172	12.9	238	13.7	
Central	527	17.2	214	16.1	313	18.0	
Southern	466	15.2	157	11.8	309	17.8	
Gao-Ping	489	15.9	216	16.2	273	15.7	
Eastern	119	3.9	56	4.2	63	3.6	

**Table 2 medicina-58-00820-t002:** Analysis of the characteristics of domestic and imported cases of amebiasis in Taiwan from 2011 and 2020 by year.

Variables	Year										*p*
	2011	2012	2013	2014	2015	2016	2017	2018	2019	2020	
Cases											
Domestic	120	86	88	103	155	148	188	164	157	122	<0.001
Imported	135	172	183	197	196	167	191	171	195	128	
Sex											
Male	151	112	139	155	187	175	239	197	194	158	<0.001
Female	104	146	132	145	164	140	140	138	158	92	
Age group											
<20	1	3	4	2	2	2	1	0	7	4	0.003
20–29	101	86	107	100	136	124	112	114	120	75	
30–39	84	105	100	127	130	127	145	134	127	106	
40–49	30	27	44	44	45	33	57	50	46	32	
50–59	17	20	9	17	19	11	37	19	25	12	
≥60	22	17	7	10	19	18	27	18	27	21	
Season											
Spring	69	67	70	66	94	72	92	87	91	65	0.467
Summer	53	73	84	76	91	93	100	78	78	59	
Fall	72	62	61	93	85	82	90	102	94	67	
Winter	61	56	56	65	81	68	97	68	89	59	
Residency											
Taipei area	78	103	95	102	129	104	118	117	136	73	<0.001
Northern area	37	38	31	50	36	40	48	47	51	32	
Central area	41	45	39	40	57	58	73	64	45	65	
Southern area	45	39	45	44	55	58	53	49	37	41	
Kao-Ping area	40	25	40	48	61	49	78	47	73	28	
Eastern area	14	8	21	16	13	6	9	11	10	11	

**Table 3 medicina-58-00820-t003:** Association between season and gender, age groups, and region of residence from a survey of domestic and imported cases of amebiasis between 2011 and 2020 in Taiwan.

Variables	Season				*p*
	Spring	Summer	Fall	Winter	
Sex					
Male	441	423	442	401	0.448
Female	332	362	366	299	
Age group					
<20	5	4	9	8	0.737
20–29	285	274	282	234	
30–39	298	297	315	275	
40–49	89	110	108	101	
50–59	43	55	51	37	
≥60	53	45	43	45	
Residency					
Taipei area	257	271	270	257	0.163
Northern	98	109	119	84	
Central	129	137	136	125	
Southern	148	99	117	102	
Kao-Ping	112	138	136	103	
Eastern	29	31	30	29	

**Table 4 medicina-58-00820-t004:** Association between region of residence and gender and age groups from a survey of domestic and imported cases of amebiasis between 2011 and 2020 in Taiwan.

Variables	Region of Residence						*p*
	Taipei Area	Northern Area	Central Area	Southern Area	Kao-Ping Area	Eastern Area	
Sex							
Male	569	215	322	225	310	66	<0.001
Female	486	195	205	241	179	53	
Age group							
<20	6	3	8	3	4	2	<0.001
20–29	316	168	203	195	164	29	
30–39	416	165	220	170	171	43	
40–49	181	47	46	33	82	19	
50–59	75	14	21	29	37	10	
≥60	61	13	29	36	31	16	

**Table 5 medicina-58-00820-t005:** Association between gender and age groups from a survey of domestic and imported cases of amebiasis between 2011 and 2020 in Taiwan.

Variables	Sex		*p*
	Male	Female	
Age group			
<20	19	7	<0.001
20–29	532	543 ^a^	
30–39	595	590 ^b^	
40–49	266 ^c^	142	
50–59	162 ^d^	24	
≥60	133 ^e^	53	

^a^: Cases with age 20–29 years old (OR = 1.470, 95% CI = 1.266–1.706 (*p* < 0.001)) in females compared with males; ^b^: Cases with age 30–39 years old (OR = 1.434, 95% CI = 1.239–1.660 (*p* < 0.001)) in females compared with males ^c^: Cases with age 40–49 years old (OR = 1.582, 95% CI = 1.273–1.966 (*p* < 0.001)) in males compared with females ^d^: Cases with age 50–59 years old (OR = 5.833, 95% CI = 3.776–9.010 (*p* < 0.001)) in males compared with females ^e^: Cases with age ≥60 years old OR = 2.082, 95% CI = 1.502–2.887 (*p* < 0.001)) in males compared with females.

**Table 6 medicina-58-00820-t006:** The number of imported amebiasis cases reported by country, 2011–2020.

Country	Year									
	2011	2012	2013	2014	2015	2016	2017	2018	2019	2020
Asia (N = 1716)										
China	7	5	6	3	4	3	2	4	6	
Korea						1		1	1	
Japan				1		1		2		1
Philippines	8	19	12	22	20	17	30	18	24	13
Indonesia	106	141	153	164	159	127	148	122	155	110
Vietnam	6	2	5	3	6	6	4	13	4	2
Myanmar		1	1	1	1					1
Cambodia	1					2	1	1		
Malaysia					1	1				
Thailand	3	3	4	1		2	4	3	4	1
Maldives	1									
India	2				2	4	1	1	1	
Oceania (N = 5)										
Australia					1	1				
Giribas					1					
Tuvalu								2		
America (N = 3)										
USA				1						
Panama			1							
Colombia							1			
Europe (N = 2)										
U. K.		1								
France						1				
Miss. Data (N = 9)	1	0	1	1	1	1	0	4	0	0

**Table 7 medicina-58-00820-t007:** Travel destinations of 1695 imported cases of *E. histolytica* infection in Taiwan between 2011 and 2020.

Country of Destination	No. Cases	No. of Air Passengers (100,000)	RR
China	40	271.8	Reference
Vietnam	51	21.7	16.0
Thailand	25	18.2	9.3
Philippines	183	20.5	60.7
Indonesia	1385	17.3	544.0
India	11	2.9	25.8

Note: Only countries with at least 10 cases of infection with *E. histolytica* were listed. RR: relative risk.

**Table 8 medicina-58-00820-t008:** Analysis of deaths due to amebiasis in Taiwan from 2011 to 2020.

Variables	Year *					
	Overall	2011	2013	2014	2015	2018
Male	6	1	1	1	2	1
<20	-	-	-	-	-	-
20–44	2	-	-	1-	1	-
45–64	2	1	1	-	-	-
≥65	2	-	-	-	1	1
Female	2	1	-	-	1	-
<20	-	-	-	-	-	-
20–44	-	-	-	-	-	-
45–64	1	1	-	-	-	-
≥65	1	-	-	-	1	-

* years 2012, 2016, 2017, 2019, and 2020 had no deaths due to amebiasis.

**Table 9 medicina-58-00820-t009:** Association between air pollutant factors and amebiasis cases by multiple linear regression analysis.

Variables	Non-Standardization Coefficient		*p*
	B Value	Standard Error	
TSP (μg/m^3^)	−0.099	0.087	0.259
PM 2.5 (μg/m^3^)	−0.320	0.279	0.254
SO_2_ (ppb)	0.607	1.761	0.731
CO (ppm)	−63.818	36.994	0.088
NO_2_ (ppb)	2.569	1.074	0.019
O_3_ (ppb)	0.294	0.109	0.008

R^2^ = 0.145; F = 2.519 (*p* value: 0.027); df = (6, 89). N = 96.

**Table 10 medicina-58-00820-t010:** Association between climate factors and amebiasis cases by multiple linear regression analysis.

Variables	Non-Standardization Coefficient		*p*
	B Value	Standard Error	
Temperature (°C)	1.096	0.542	0.046
Precipitation (mm)	0.001	0.010	0.890
Relative humidity (%)	−0.524	0.392	0.184
Mean Pressure (hPa)	0.349	0.451	0.441
Number of Days with Precipitation ≥0.1 mm (day)	0.137	0.519	0.792
Sunshine Duration (hr)	−0.038	0.041	0.353

R^2^ = 0.073; F = 1.165 (*p* value: 0.332); df = (6, 89). N = 96.

## Data Availability

Not applicable.

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
