# Peer review of "The Epidemiology of Entamoeba histolytica Infection and Its Associated Risk Factors among Domestic and Imported Patients in Taiwan during the 2011–2020 Period"

_medicina, 2022, doi:10.3390/medicina58060820_

Round 1

Reviewer 1 Report

The manuscript addresses an interesting public health problem that not only affects countries traditionally considered endemic for E. histolytica/E dispar infection, but also other countries such as Taiwan, which shows a relatively low incidence of amebiasis in its country. population, the article clearly shows how migration to countries endemic for amebiasis has affected the increase in these figures in recent years.

  Another aspect that I want to highlight is the consideration of evaluating other risks that are rarely considered in the literature, such as environmental risks, which in this case is novel and interesting.

Author Response

Dear the reviewers,                                         June 15, 2022

  We resubmitted the manuscript entitled “Epidemiology and Risk Factors Associated with Entamoeba Histolytica Infection among Domestic and Imported Patients in Taiwan during the Period 2011-2020” to the Journal after amendments made based on reviewers comments. We have carefully revised our manuscript according to reviewers’ critiques and suggestions. We marked amendments in yellow font in the manuscript for clarity purpose. Our specific responses to reviewers’ comments are as follows.

Reviewer 1

The manuscript addresses an interesting public health problem that not only affects countries traditionally considered endemic for E. histolytica/E dispar infection, but also other countries such as Taiwan, which shows a relatively low incidence of amebiasis in its country. population, the article clearly shows how migration to countries endemic for amebiasis has affected the increase in these figures in recent years.

Another aspect that I want to highlight is the consideration of evaluating other risks that are rarely considered in the literature, such as environmental risks, which in this case is novel and interesting.

Response:

Thanks the reviewer comment. The authors sincerely thank the reviewers for their positive views on this study.

Hopefully, our revised manuscript could fulfill your scientific requirements for publication.

Sincerely yours,

Chia-Peng Yu, Ph.D. (the corresponding author)

School of Public Health,

National Defense Medical Center

No.161 Sec. 6, Minquan E. Rd., Neihu Dist., Taipei 114, Taiwan, Republic of China,

Tel: +886-2-87923311 ext. 16791, Fax: +886-2-87924379,

Reviewer 2 Report

To editors and reviewers
Epidemiology and Risk Factors Associated with Entamoeba Histolytica Infection among Domestic and Imported Patients in Taiwan during the Period 2011-2020
- This is a very interesting manuscript that can be considered for publication in MEDICINA. The manuscript is appropriate with aims and scope of journal. 
- I suggested some revisions below and after revisions the manuscript can be published.
1) Some citation and references are not precise as MDPI format. Please check and revise.

2) Abstract for a original research should be structured.

3) The limitation part should be independent and clear. Please revise

4) The conclusion is an independent part.

Sincerely

Author Response

Dear the reviewers,                                         June 15, 2022

  We resubmitted the manuscript entitled “Epidemiology and Risk Factors Associated with Entamoeba Histolytica Infection among Domestic and Imported Patients in Taiwan during the Period 2011-2020” to the Journal after amendments made based on reviewers comments. We have carefully revised our manuscript according to reviewers’ critiques and suggestions. We marked amendments in yellow font in the manuscript for clarity purpose. Our specific responses to reviewers’ comments are as follows.

Reviewer 2

Epidemiology and Risk Factors Associated with Entamoeba Histolytica Infection among Domestic and Imported Patients in Taiwan during the Period 2011-2020

- This is a very interesting manuscript that can be considered for publication in MEDICINA. The manuscript is appropriate with aims and scope of journal.

- I suggested some revisions below and after revisions the manuscript can be published.

1) Some citation and references are not precise as MDPI format. Please check and revise.

Response:

Thanks the reviewer comment. The authors have been checked and revised citation and references of the Manuscript as MDPI format. Please see line 375-470.

2) Abstract for a original research should be structured.

Response:

Thanks the reviewer comment. The authors have been structured of abstract in the revised manuscript. Please see line 19-40.

3) The limitation part should be independent and clear. Please revise

Response:

The authors have added the sentence as “5. Limitations” in revised manuscript. And the paragraph of limitation part is clear in the revised manuscript. Please see line 334-351.

4) The conclusion is an independent part.

Response:

The authors have added the sentence as “6. Conclusions” in revised manuscript. Please see line 352-360.

Hopefully, our revised manuscript could fulfill your scientific requirements for publication.

Sincerely yours,

Chia-Peng Yu, Ph.D. (the corresponding author)

School of Public Health,

National Defense Medical Center

No.161 Sec. 6, Minquan E. Rd., Neihu Dist., Taipei 114, Taiwan, Republic of China,

Tel: +886-2-87923311 ext. 16791, Fax: +886-2-87924379,

This manuscript is a resubmission of an earlier submission. The following is a list of the peer review reports and author responses from that submission.